# Nurturing 21st century physician knowledge, skills and attitudes with medical home innovations: the Wright Center for Graduate Medical Education teaching health center curriculum experience

Linda Thomas-Hemak[1], Ghanshyam Palamaner Subash Shantha[1], Lakshmi Rani Gollamudi[1], Jignesh Sheth[1], Brian Ebersole[1], Katlyn J. Gardner[1], Julie Nardella[1], Meaghan P. Ruddy[1] and Lauren Meade[1,2]

[1] The Wright Center for Graduate Medical Education, Internal Medicine, Scranton, PA, USA
[2] Tufts Medical School, Baystate Health, Springfield, MA, USA

Corresponding author
Linda Thomas-Hemak,
thomasl@thewrightcenter.org

## ABSTRACT

**Purpose.** The effect of patient centered medical home (PCMH) curriculum interventions on residents' self-reported and demonstrated knowledge, skills and attitudes in PCMH competency arenas (KSA) is lacking in the literature. This study aimed to assess the impact of PCMH curricular innovations on the KSA of Internal Medicine residents.

**Methods.** Twenty four (24) Internal Medicine residents—12 Traditional (TR) track residents and 12 Teaching Health Center (THC) track residents—began training in Academic Year (AY) 2011 at the Wright Center for Graduate Medical Education (WCGME). They were followed through AY2013, covering three years of training. PCMH curricular innovations were focally applied July 2011 until May 2012 to THC residents. These curricular innovations were spread program-wide in May 2012. Semi-annual, validated PCMH Clinician Assessments assessing KSA were started in AY2011 and were completed by all residents.

**Results.** Mean KSA scores of TR residents were similar to those of THC residents at baseline for all PCMH competencies. In May 2012, mean scores of THC residents were significantly higher than TR residents for most KSA. After program-wide implementation of PCMH innovations, mean scores of TR residents for all KSA improved and most became equalized to those of THC residents. Globally improved KSA scores of THC and TR residents were maintained through May 2014, with the majority of improvements above baseline and reaching statistical significance.

**Conclusions.** PCMH curricular innovations inspired by Health Resources and Services Administration (HRSA's) Teaching Health Center funded residency program expansion quickly and consistently improved the KSA of Internal Medicine residents.

## INTRODUCTION

The 21st century marks a period of dramatic shifts in health care paradigms in the United States. Health care costs in the United States have grown exponentially (*Martin et al., 2011*), but are not paralleled by improvements in health care delivery efficiencies, public health outcomes, physician skill development or patient satisfaction with health care experiences (*McGlynn et al., 2003*; *Wilensky & Berwick, 2014*). The patient-centered medical home (PCMH) shows promise as a quality, team-based and population-health focused model of innovative primary care delivery that may potentiate valuable enhancement in health care delivery with a reduction in costs and improved health outcomes (*Agency for Healthcare Research and Quality, 2011*; *Scholle et al., 2010*; *Stange et al., 2010*). PCMH enthusiasm has prompted many professional organizations, stakeholders, and policy-makers (*American Academy of Family Physicians, 2014*) to work at national, state, and local levels to ensure integration of this innovative strategy in primary care practices and promote the pursuit of National Committee for Quality Assurance (NCQA) PCMH certification. Ongoing systems of measurement and process changes in support of the implementation of high PCMH standards can enhance quality, cost-effectiveness and outcomes-focused care (*Bitton, Martin & Landon, 2010*; *Agency for Healthcare Research and Quality, 2014*).

Authentic transformation of health care delivery models requires a change in the skill sets of primary care providers (*Bowen et al., 2010*; *Johnson et al., 2010*; *Sinsky et al., 2013*; *Jortberg et al., 2014*; *Dickinson, 2010*). In a study by Kaiser Permanente, skill sets of routine office-based competencies including chronic disease management, care coordination, care continuity, familiarity with team-based care models, clinical information technology, leadership and management skills, and systems thinking were reported as deficient in the newly-trained physician workforce (*Crosson et al., 2011*). In response, various components of the Affordable Care Act legislation have aimed to address the national shortage and maldistribution of physicians in order to inspire the production of new skill sets and to reduce related health disparities. One legislative example is the Teaching Health Center Graduate Medical Education (THCGME) Program implemented by the Health Resources and Services Administration (HRSA). In 2011, THCGME pioneering grantees included nine Family Medicine, one Dental and one Internal Medicine residencies (*United States Department of Health and Human Services, 2013*; *Nutting et al., 2007*). The Wright Center for Graduate Medical Education (WCGME) is the sponsoring institution for the ACGME-accredited Internal Medicine residency in the initial cohort of THCGME programs.

WCGME sought to align educational processes with reform mandates, THCGME funding intent and local community need. The resulting comprehensive curricular redesign increased training time in ambulatory care settings, including Federally Qualified

Health Center (FQHC) exposure, provided focused PCMH and leadership didactics, and aligned engagement in ambulatory care-based continuous quality improvement projects. This study provides evidence of the changes in the KSA of residents in support of the teaching health center-based curriculum. This curriculum aligns graduate medical education with the complexities of health care reform and patient care needs.

## MATERIALS AND METHODS

After approval of this prospective cohort study by the Institutional Review Board of the Wright Center, informed consent was obtained from each participating resident. Measures were implemented to ensure the anonymity of resident participants and the assessment data they provided throughout the study duration. Since AY 2011, all Internal Medicine residents completed semi-annual PCMH Clinician Assessments, as well as other data collection tools.

### Setting

PCMH-based didactics enhancements and resident engagement in reflective continuous quality improvement (CQI) within ambulatory venues with or pursuing National Committee for Quality Assurance (NCQA) PCMH certification were focally applied to 2011 Teaching Health Center (THC) track residents from training initiation in July 2011. In May 2012, these changes were spread program-wide. Innovations specifically applied to THC residents from the start of training also included increased block continuity clinic exposure split between a Wright Center for Primary Care (WCPC) and a proximal FQHC, with an intentional ratio of 50% ambulatory and 50% hospital-based rotations. In contrast, Traditional (TR) track residents continued with the historical half day/week continuity clinic exclusively at WCPC and had 35% ambulatory and 65% hospital-based rotation exposure. Continuity ambulatory and hospital rotational exposure models remained different between 2011 THC and TR training tracks of study participants for the entire three years of their training.

### Participating residents

WCGME recruited 12 THC and 12 TR residents to start Internal Medicine training in AY2011. Those reporting definitive interest in ambulatory medicine were assigned to the THC track, and those uncertain or interested in hospitalists or specialty careers were randomly assigned to the THC or TR track. One 2011 THC resident left the program in January 2014, and one 2011 THC resident had a 3 month training delay and graduated off cycle in September 2014. During the study period, 10 and 12 TR residents, who started training in 2009 and 2010 respectively, completed training and graduated (2009 and 2010 TR Graduating Seniors).

### PCMH curricular innovations

A detailed explanation of the curriculum is provided at http://www.thewrightcenter. org/graduate-medical-education/residency/allopathic-internal-medicine/. In addition to increased FQHC and expanded block-based continuity clinic exposure, THC residents

were exposed to enhanced PCMH-based didactics, clinic team huddles and CQI ambulatory projects. Weekly team huddles identified care delivery deficits for which THC residents helped design and implement team-based remediations. As part of the Pennsylvania Academy of Family Practice's Improving Performance in Practice Residency Program/Community Health Center Collaborative, residents engaged in team-based CQI projects. They also participated in monthly team calls and quarterly learning sessions. The didactic curriculum was augmented to include weekly PCMH-based interactive sessions for THC track residents including topics on PCMH delivery model, Quality Improvement fundamentals, Population Management, EMR Meaningful Use, Team Skills and Leadership.

These interventions were first implemented in the WCGME in July 2011. From onset until May 2012, these didactic interventions were focally applied only to THC residents. Because THC residents consistently outperformed TR cohort colleagues and TR 2009 Graduating Senior Residents in most KSA by end of AY2011, PCMH curricular innovations were spread program-wide starting in May 2012. Future AY recruits were all integrated into the new, balanced THC clinical exposure model with block ambulatory time spilt between a Wright Center for Primary Care and proximal FQHC. Mandatory clinical training curricular enhancements in Women's Health, Oral Hygiene and Primary Care Psychiatry were added in AY2012 only for THC residents.

## Assessment details

Demographic details including age, gender, country of origin, prior residency training, completion of other advanced degrees such as masters of public health (MPH) or doctoral degrees, prior US clinical experience, prior research experience and year of medical school graduation were compared between 2011 THC and TR residents at baseline. Semi-annual November and May validated PCMH Clinician Assessments were completed by all Internal Medicine residents at WCGME since 2011. This PCMH Clinician Assessment assesses nine KSA: (1) Team Approach: the utilization of physician-led team-based health care delivery that has all members practicing at the highest level of their license to engage patients and optimize care; (2) Information System Support: encompasses meaningful use of electronic medical records as well as accurate, timely and effective flow of information between health care delivery stakeholders that assures ready access to key data on individual patients and the overall population; (3) Self-Management Support: acknowledgement of the patients' central role in their care, one that fosters a sense of responsibility for their own health and resiliency. Using a collaborative approach, providers and patients work together to define problems, set priorities, establish obtainable goals, create treatment plans and solve problems along the way; (4) Use of Guidelines: clinical decision making to promote utilization of evidence in facilitating shared-decision making between the care team and the patients; (5) Quality Improvement: systematic approach to the reduction and elimination of waste, rework and losses in production encouraged by reflective practice for improvement and just-in-time alterations of health care delivery; (6) Population Management: management of the health outcomes of a group

of individuals, including the distribution of such outcomes within the group. It is an approach to health that aims to improve the health of an entire population served; (7) Care Coordination: deliberate organization of patient care activities across the care continuum to reduce duplicative efforts and achieve better outcomes; (8) Patient Centered Care: providing care that is respectful of and responsive to individual patient preferences, needs, and values, and ensuring that patient values guide all clinical decisions; and (9) Treatment of Mental Health Issues: integration of screening and mental health care referral/provider services into the primary care processes realizing it is the first line of defense for the diagnosis, treatment and prevention of mental health issues. Each assessment question has self-rated score options from 1–5, with 1 denoting limited and 5 denoting superior competency. This survey tool was developed and used by *Jortberg et al. (2014)* to assess the KSA of family medicine residents. A copy of the survey tool can be found in File S1 and also by following the link http://pcmhelearning.com/theme/intervision_custom/pdf/PCMH.CA.pdf. As described by *Jortberg et al. (2014)*, a previous version of this tool has been used to assess clinician reported use of elements of chronic care model in primary care setting (*Nutting et al., 2007*; *Nease et al., 2008*). *Jortberg et al. (2014)* added detailed components to KSA in the previous version and used it in their study on family medicine residents. We used the same survey tool from the paper by *Jortberg et al. (2014)* and employed it to assess the KSA of our IM residents. The Cronbach's alpha assessing reliability for each of the PCMH competencies in the survey tool were > 0.8.

The analysis focused on the 2011 cohort of THC and TR Residents (12 in each group) as a comparative platform to assess historical and new curricular innovations. In total, 2011 THC and TR residents completed six assessments from November 2011 through May 2014. We compared four assessments longitudinally within and between the 2011 THC and TR tracks from the following time points: (1) Baseline Assessment (November 2011); (2) End of first training year (May 2012); (3) Second year of training (November 2012); (4) Graduation assessment near training completion (May 2014).

The scores of the 2011 THC and TR residents in May 2012 and near graduation in May 2014 were also compared with the scores of the 2009 and 2010 TR graduating seniors in May of their respective graduation years. The rationale for this analysis is that 2009 and 2010 TR graduating seniors were trained prior to implementation of THC PCMH innovations, noting that 2009 TR graduates had no direct exposure to curricular innovations described and 2010 TR graduates had one year of exposure after the program-wide rollout of the curricular innovations. The 2009 and 2010 TR trainees completed two and four PCMH Clinician Assessments respectively.

## STATISTICAL ANALYSIS

Data was expressed as mean ± standard deviation for continuous variables and as a number (%) for categorical variables. Data distribution normality was assessed using the Kolmogorov–Smirnov test. Considering the small sample sizes, our data were predominantly non-normally distributed, so non-parametric tests were used for all comparisons. Due to the non-parametrically distributed data, continuous variables

Table 1 **Baseline demographic characteristics of 2011 THC and 2011 TR residents.** Baseline demographic comparisons between THC and TR residents.

| | 2011 THC residents (*n* = 12) | 2011 TR residents (*n* = 12) | *P*-value |
|---|---|---|---|
| Age (yrs) | 34 [5.5] | 31 [3] | 0.008 |
| Males—*n* (%) | 6 (50) | 7 (58) | 1.000 |
| Country of origin | | | 0.515 |
| India—*n* (%) | 6 (50) | 7 (58) | |
| Egypt—*n* (%) | 3 (25) | 0 (0) | |
| USA—*n* (%) | 1 (8) | 2 (17) | |
| China—*n* (%) | 1 (8) | 2 (17) | |
| Syria—*n* (%) | 0 (0) | 1 (8) | |
| Libya—*n* (%) | 1 (8) | 0 (0) | |
| Years since medical school grad. (yrs) | 8 (4) | 4 (3.5) | 0.001 |
| Prior residency training—*n* (%) | 10 (83) | 7 (58) | 0.185 |
| Advanced degrees—*n* (%) | 3 (25) | 5 (42) | 0.333 |
| Prior clinical experience–*n* (%) | 12 (100) | 11 (92) | 0.500 |
| Prior research experience—*n* (%) | 7 (58) | 8 (67) | 0.500 |

were compared between groups using the Wilcoxon rank sum test. Categorical variables were compared between groups using a chi-squared test. Mean group based scores for KSA assessed were compared longitudinally for both 2011 THC and TR residents over 3 years of training at the above-defined time points, with comparisons to just prior and baseline scores. Between groups, cross-sectional, comparisons of mean individual and global KSA scores at the defined time points were done for 2011 THC and TR cohort residents. Similar cross sectional comparisons between groups for each 2011 cohort's mean individual and global scores in May of their internship and graduation year were also done near graduation in May surveys of 2009 and 2010 TR residents in their respective graduation year. All analyses were performed using STATA 11 statistical software. $P < 0.05$ was considered statistically significant.

## RESULTS

Response rates for all assessments were 100%. Demographic comparisons detailed in Table 1 showed 2011 THC residents were older and completed medical school earlier than TR resident peers (Table 1). Gender distribution, country of origin, prior residency training, advanced degrees, prior US clinical and research experiences were similar (Table 1).

### Longitudinal within groups 2011 THC and TR residents comparison

Compared to the November 2011 baseline, the assessments of May 2012 THC residents showed mean score improvement for all KSA except Patient Centered Care. THC mean scores for all individual and global KSA improved from baseline on the May 2014 assessments, with majority of improvements reaching statistical significance (Table 2).

**Table 2 Longitudinal KSA 2011 THC residents comparison to baseline.** Longitudinal KSA comparisons for THC residents at the pre-defined time points compared to their baseline.

| Competency | Nov 2011 n = 12 | May 2012 n = 12 | P value | Nov 2012 n = 12 | P-value | May 2014 n = 11 | P value |
|---|---|---|---|---|---|---|---|
| Care coordination | 4.0 (3.5–4.5) | 4.7 (4.3–5.0) | 0.039 | 4.3 (3.9–4.7) | 0.106 | 4.7 (4.4–4.9) | 0.031 |
| Info system support | 3.5 (3.2–3.8) | 4.4 (4.0–4.7) | 0.001 | 3.9 (3.7–4.2) | 0.113 | 4.5 (4.2–4.8) | 0.001 |
| Patient centered care | 4.2 (3.7–4.7) | 4.1 (3.7–4.5) | 0.283 | 4.2 (3.6–4.8) | 0.558 | 4.6 (4.3–4.9) | 0.092 |
| Population management | 3.8 (3.4–4.2) | 4.6 (4.1–4.9) | 0.001 | 4.5 (4.1–4.9) | 0.022 | 4.6 (4.3–4.9) | 0.028 |
| Quality improvement | 4.0 (3.6–4.5) | 4.3 (3.9–4.7) | 0.094 | 4.5 (4.2–4.9) | 0.039 | 4.6 (4.3–4.9) | 0.037 |
| Self-man support | 4.1 (3.5–4.6) | 4.5 (4.1–4.8) | 0.021 | 4.5 (4.3–4.9) | 0.047 | 4.5 (4.2–4.8) | 0.057 |
| Team approach | 3.8 (3.4–4.3) | 4.6 (4.3–5.0) | 0.001 | 4.5 (4.2–4.9) | 0.001 | 4.4 (4.0–4.7) | 0.016 |
| Mental health Tx | 4.4 (3.8–4.9) | 4.9 (4.5–5.0) | 0.041 | 4.7 (4.4–5.0) | 0.296 | 4.6 (4.3–4.9) | 0.314 |
| Use of guidelines | 4.2 (3.7–4.7) | 4.5 (4.2–4.9) | 0.089 | 4.4 (4.1–4.8) | 0.217 | 4.6 (4.3–4.9) | 0.100 |

**Table 3 Longitudinal KSA2011 TR residents comparison to baseline.** Longitudinal KSA comparisons for TR residents at pre-defined time points compared to their baseline.

| Competency | Nov 2011 n = 12 | May 2012 n = 12 | P value | Nov 2012 n = 12 | P-value | May 2014 n = 12 | P value |
|---|---|---|---|---|---|---|---|
| Care coordination | 4.1 (3.5–4.7) | 4.1 (3.6–4.5) | 0.833 | 4.2 (3.7–4.6) | 0.625 | 4.3 (3.7–4.9) | 0.225 |
| Info system support | 3.9 (3.0–4.0) | 4.2 (3.8–4.6) | 0.068 | 4.2 (3.6–4.7) | 0.109 | 4.3 (3.6–4.9) | 0.053 |
| Patient centered care | 4.3 (3.8–4.7) | 4.4 (4.0–4.7) | 0.532 | 4.2 (3.8–4.6) | 0.812 | 4.4 (3.5–4.9) | 0.719 |
| Population management | 3.6 (3.0–4.3) | 4.0 (3.5–4.3) | 0.094 | 4.1 (3.6–4.6) | 0.077 | 4.2 (3.5–4.8) | 0.021 |
| Quality improvement | 3.8 (3.1–4.5) | 4.1 (3.5–4.5) | 0.561 | 4.1 (3.5–4.8) | 0.104 | 4.3 (3.7–4.8) | 0.033 |
| Self-man support | 4.0 (3.5–4.3) | 4.2 (3.5–4.5) | 0.735 | 4.3 (4.0–4.7) | 0.113 | 4.5 (4.0–5.0) | 0.017 |
| Team approach | 3.8 (3.5–4.5) | 4.3 (4.0–4.9) | 0.199 | 4.0 (3.5–4.6) | 0.784 | 4.4 (3.9–5.0) | 0.038 |
| Mental health Tx | 4.0 (3.3–4.7) | 4.4 (4.0–4.8) | 0.187 | 4.4 (4.0–4.8) | 0.172 | 4.5 (4.1–5.0) | 0.036 |
| Use of guidelines | 4.2 (3.7–4.5) | 4.4 (4.0–4.8) | 0.116 | 4.2 (3.7–4.7) | 0.978 | 4.5 (4.0–4.9) | 0.136 |

Unlike their THC peers, the mean scores of TR residents in May 2012 remained similar to the November 2011 baseline without improvement for most KSA. The May 2014 TR scores did show significant improvement in all KSA, with improvements above baseline for Team Approach, Self-Management Support, Quality Improvement, Population Management and Treatment of Mental Health Issues reaching statistical significance (Table 3).

## Comparison between 2011 THC and TR residents groups

Baseline assessments showed similar performance in all KSA when 2011 THC residents were compared to 2011 TR peers (Table 4). May 2012 assessments showed 2011 THC residents consistently scored higher in most and global KSA than their TR peers and 2009 TR graduating seniors (Table 4; Fig. 1 and Table S1; Fig. S1). Because of the notable May 2012 performance improvement of the 2011 THC residents in most KSA above baseline and the superior performance above 2011 TR peers and 2009 TR graduating seniors, the
**Table 4  KSA compared between THC residents (*n* = 12) and TR Residents (*n* = 12).** KSA comparisons between TR and THC residents at pre-defined time points.

| | November 2011 | | | May 2012 | | | November 2012 | | | May 2014 | | |
|---|---|---|---|---|---|---|---|---|---|---|---|---|
| | THC | TR | *P*-value | THC | TR | *P*-value | THC | TR | *P*-value | THC | TR | *P*-value |
| Care coordination | 4.0 (3.5–4.5) | 4.1 (3.5–4.7) | 0.428 | 4.7 (4.3–5.0) | 4.1 (3.6–4.5) | 0.025 | 4.3 (3.9–4.7) | 4.2 (3.7–4.6) | 0.660 | 4.7 (4.4–4.9) | 4.3 (3.7–4.9) | 0.268 |
| Info system support | 3.5 (3.2–3.8) | 3.9 (3.0–4.0) | 0.107 | 4.4 (4.0–4.7) | 4.2 (3.8–4.6) | 0.177 | 3.9 (3.7–4.2) | 4.2 (3.6–4.7) | 0.082 | 4.5 (4.2–4.8) | 4.3 (3.6–4.9) | 0.811 |
| Patient centered care | 4.2 (3.7–4.7) | 4.3 (3.8–4.7) | 0.382 | 4.1 (3.7–4.5) | 4.4 (4.0–4.7) | 0.392 | 4.2 (3.6–4.8) | 4.2 (3.8–4.6) | 0.537 | 4.6 (4.3–4.9) | 4.4 (3.5–4.9) | 0.515 |
| Population management | 3.8 (3.4–4.2) | 3.6 (3.0–4.3) | 0.173 | 4.6 (4.1–4.9) | 4.0 (3.5–4.3) | 0.036 | 4.5 (4.1–4.9) | 4.1 (3.6–4.6) | 0.115 | 4.6 (4.3–4.9) | 4.2 (3.5–4.8) | 0.297 |
| Quality improvement | 4.0 (3.6–4.5) | 3.8 (3.1–4.5) | 0.225 | 4.3 (3.9–4.7) | 4.1 (3.5–4.5) | 0.091 | 4.5 (4.2–4.9) | 4.1 (3.5–4.8) | 0.210 | 4.6 (4.3–4.9) | 4.3 (3.7–4.8) | 0.319 |
| Self-man support | 4.1 (3.5–4.6) | 4.0 (3.5–4.3) | 0.491 | 4.5 (4.1–4.8) | 4.2 (3.5–4.5) | 0.047 | 4.5 (4.3–4.9) | 4.3 (4.0–4.7) | 0.307 | 4.5 (4.2–4.8) | 4.5 (4.0–5.0) | 0.372 |
| Team approach | 3.8 (3.4–4.3) | 3.8 (3.5–4.5) | 0.651 | 4.6 (4.3–5.0) | 4.3 (4.0–4.9) | 0.039 | 4.5 (4.2–4.9) | 4.0 (3.5–4.6) | 0.137 | 4.4 (4.0–4.7) | 4.4 (3.9–5.0) | 0.382 |
| Mental health Tx | 4.4 (3.8–4.9) | 4.0 (3.3–4.7) | 0.251 | 4.9 (4.5–5.0) | 4.4 (4.0–4.8) | 0.163 | 4.7 (4.4–5.1) | 4.4 (4.0–4.8) | 0.126 | 4.6 (4.3–4.9) | 4.5 (4.1–5.0) | 0.560 |
| Use of guidelines | 4.2 (3.7–4.7) | 4.2 (3.7–4.5) | 0.815 | 4.5 (4.2–4.9) | 4.4 (4.0–4.8) | 0.711 | 4.4 (4.1–4.8) | 4.2 (3.7–4.7) | 0.534 | 4.6 (4.3–4.9) | 4.5 (4.0–4.9) | 0.502 |

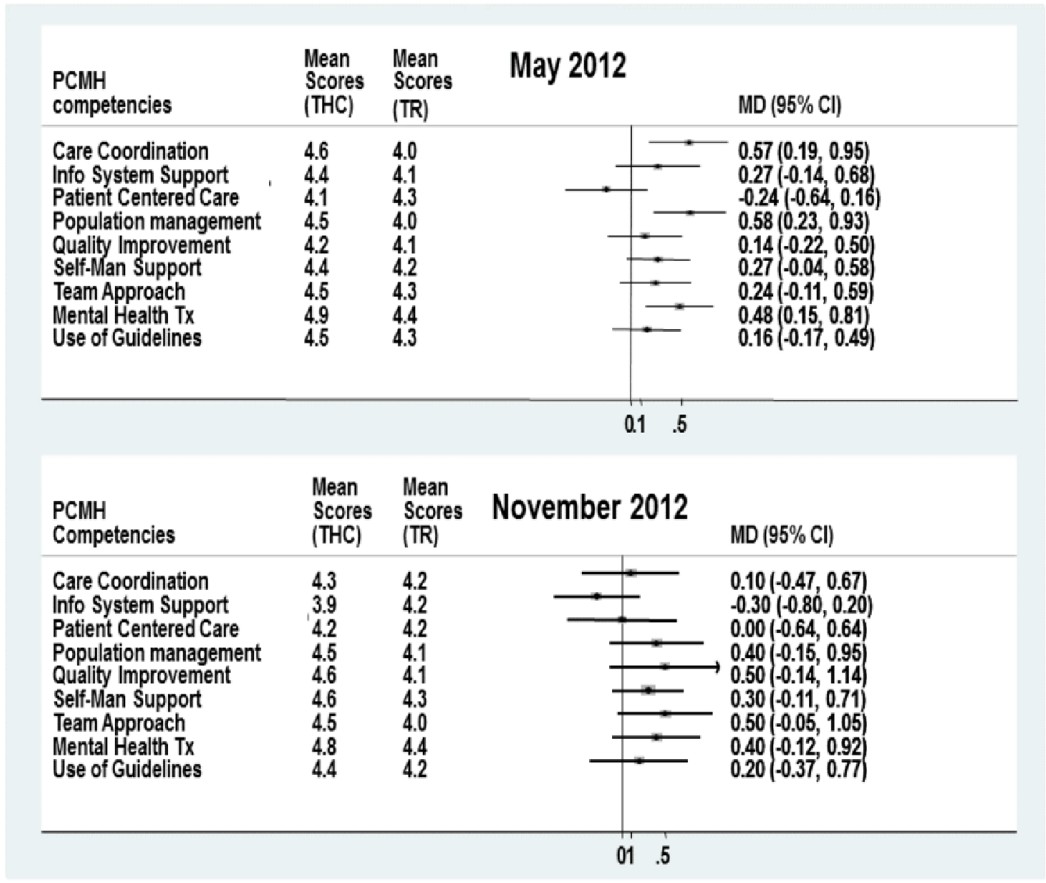

**Figure 1** KSA comparison of 2011 THC and 2011 TR Residents May 2012 and Nov 2012.

PCMH curricular innovations were spread program-wide. However, to preserve continuity exposure, the redesigned training model of increased block-based ambulatory continuity split between a WCPC and proximal FQHC was not spread program-wide at this time as program leadership opted for a phased roll out for this training model for all new interns, starting in AY2012.

The November 2012 KSA assessment completed six months after program-wide spread of PCMH curriculum innovations revealed that 2011 TR residents showed global, but not statistically significant improvement in KSA from baseline. However, TR mean scores did equalize for all KSA, with the THC colleagues at this time, and remained comparable to that of their THC peers through May 2014. (Fig. 1 and Table 4)

The 2011 THC residents mean scores for most KSA in May 2014 were consistently better for most residents and global KSA than those of the 2009 TR graduating seniors in May 2012, with 5 of 9 KSA improvements reaching statistical significance (Table S2; Fig. S2). Similarly to their THC peers near-completion of their training in May 2014, 2011 TR residents scored significantly higher in 4 of 9 KSA compared to the May 2012 assessment completed by the 2009 graduating TR seniors (Table S3). Notably, both 2011 THC and TR mean scores in May 2014 were essentially similar to May 2013 scores on near graduation

May assessments completed by the 2010 TR graduating seniors (Tables S2 and S3; Figs. S3 and S4), the latter of whom did have one year of direct exposure to PCMH curriculum innovations in their senior year.

## DISCUSSION

Our study shows that multi-dimensional PCMH-based curricular innovations in the WCGME's Internal Medicine residency significantly enhanced the KSA of THC residents by the end of their internship year, and that these improvements over baseline were maintained through three years of training. Subsequent program-wide implementation of PCMH curricular innovations in May 2012 enhanced KSA in both 2011 and 2009 TR residents within a similar time frame.

The effect of PCMH innovations on patient outcomes, care delivery organizational structure and cost effectiveness have been debated in the literature (*Luck et al., 2014*; *Friedberg et al., 2014*; *Jackson et al., 2013*; *Hochman et al., 2013*). *Hochman et al. (2013)* showed that after one year of PCMH interventions, using NCQA's PCMH certification tool, the satisfaction of the residents with their experience of patient care improved significantly, with an increased perception that patients received comprehensive care at their facility. However, the effect of the PCMH curricular innovations on KSA is scant. Our study provides evidence that the intentional PCMH curricular innovations can improve residents' self-reported KSA in arenas reported as deficient in Kaiser Permanente's survey (*Crosson et al., 2011*).

Amidst all of the implemented PCMH curricular innovations and increased ambulatory care center exposure during three years of residency training, our program was sensitive to the potential distraction from the traditional Internal Medicine residency training focus that ensures broad, specialty specific medical knowledge acquisition. Prior research (*Palamaner Subash Shantha, Meade & Thomas-Hemak, 2013*) compared our 2011 THC and TR residents' first In-Training Exam (ITE) performance, an objective metric assessing comprehensive Internal Medicine medical knowledge. In this assessment, the 2011 THC residents performed similarly to their TR counterparts in all specialties except for a slight underperformance in Pulmonary/Critical Care Medicine but also showed a notably superior performance in Endocrinology ($P = 0.021$) (*Palamaner Subash Shantha, Meade & Thomas-Hemak, 2013*). Presumably due to exposure to diabetes care as a ubiquitous PCMH target population, this potential to increase Endocrinology knowledge offers promise in enhancing physician knowledge to counter our national diabetes epidemic. We realized the need for KSA to parallel objective testimonials from faculty physicians, peers, staff and patients directly observing our residents in patient care venues. In the File S1, we have detailed an analysis where we assessed the six (6) core competencies of the Accreditation Council for Graduate Medical Education (ACGME) for these 24 residents. This analysis of the ACGME core competency-based evaluations, an objective assessment to confirm enhanced skills, showed that the 2011 THC and TR residents improved their mean scores for ACGME core competencies by December 2012 and sustained improvements until completion of their training in June 2014 (Analysis S1

and Tables S4–S14). The ACGME core competencies were mapped to the KSA and several linked scores compared showed positive correlations for both 2011 THC and TR residents by December 2012. These correlations were sustained until June 2014 (Analysis S1 and Tables S4–S14). This analysis suggests that PCMH curricular innovations, in addition to improving KSA, also enhanced the residents' directly observed skill sets for effective patient care delivery in the PCMH model evidenced within an ACGME core competency framework.

Although earlier small sample studies showed enhanced patient outcomes with the PCMH model (Bidassie et al., 2014; Luck et al., 2014), a recent large study by Friedberg et al. (2014) showed limited improvement in quality and no improvement in health care utilization or cost effectiveness among multiple practices that adopted this model, suggesting further refinement is necessary in PCMH interventions (Friedberg et al., 2014). Financial sustainability of the PCMH interventions in their current form is unknown. As a result, PCMH model modifications are probable in the future. The creation of a physician workforce with deeper understanding of core PCMH principles offers the promise of making knowledgeable and adaptable physician leaders who will be prepared to lead the PCMH evolution to promote ongoing care delivery transformation in pursuit of the Institute for Healthcare Improvement's Triple Aim—improved quality, better outcomes and lower cost.

## Limitations

Our study had the advantages of a prospective design and follow-up through for the three year training cycle of 23/24 engaged 2011 THC and TR residents with a consistent 100% KSA survey response rate. Small sample size was a limiting factor precluding multivariate comparisons and control for obvious confounders, including the higher age and longer length of time since medical school graduation in the 2011 THC residents. Admittedly, our methods primarily assessed KSA, which have not been causally linked to the demonstration of these competencies in patient care delivery or actuated improvements in health outcomes. However, we did find that these improvements in KSA correlate positively with testimonials confirming an improvement in ACGME core competencies. There remains a need to further explore the relationship between enhanced KSA with enhanced patient care delivery processes and health outcomes demonstrated in the ACGME's Next Accreditation System framework for educational outcomes.

The method of recruitment based on THC track self-selection may have added selection bias as primary care focused Internal Medicine residents, who chose the THC track, may have been more oriented and committed to developing KSA. However, this selection bias seems unlikely given similar baseline assessment scores for THC and TR residents. Additionally, after program-wide implementation of PCMH-based curricular innovations, the 2011 TR residents and 2010 graduating TR seniors showed similar improvement in most KSA categories, even without the change in TR half-day continuity clinic/week and the 35% ambulatory/65% hospital-based rotational exposure.

Cross-track diffusion of didactic and CQI exposure enhancements by peer to peer education and program-wide PCMH-based curricular innovations in AY2012 may

have diluted the detectable comparative effectiveness of the innovations on training track cohorts. Finally, contemporary clinical learning environment exposure of TR residents during the half day and one-block continuity clinic rotation to the THC residents well-acclimated to ambulatory learning venues may have diluted the detectable effectiveness of curricular interventions.

## Conclusions

HRSA's THCGME funding catalyzed curricular innovations in The Wright Center's Internal Medicine Residency program. These innovations increased and enhanced PCMH-based didactics, engagement in ambulatory based CQI projects, and training time in ambulatory care centers. PCMH curricular innovations were associated with enhanced KSA and demonstrated improvements in ACGME core competencies. Improvements occurred within six months and were replicated after the curriculum spread within the same time-frame to TR residents despite their continuing in the historical training model of half day weekly continuity clinics with 35% ambulatory and 65% hospital-based training exposure.

Notable improvements in KSA above baseline and compared to prior 2009 TR graduates were sustained over three years of training. A large, multi-centered study assessing the expanded correlation of KSA with ACGME core competency-based global 360 evaluations, as well as patient care delivery process and health outcomes, is required before firm inferences can be made from our results. The ACGMES' Next Accreditation System offers a relevant, data-driven roadmap of educational and patient care outcomes for these validations. Overall, The Wright Center's Internal Medicine residency program's teaching health center curriculum experience inspires hope for a better-prepared 21st century physician workforce than previously reported by *Crosson et al. (2011)*.

### Funding

The Wright Center for Graduate Medical Education Consortium's Internal Medicine residency is funded directly through the HRSA Teaching Health Center and the Veteran Hospital GME funding, as well as through CMS funded hospitals' affiliations. The funders had no role in study design, data collection and analysis, decision to publish, or preparation of the manuscript.

### Grant Disclosures

The following grant information was disclosed by the authors:
HRSA Teaching Health Center GME.

### Competing Interests

Dr. Linda Thomas-Hemak is the President and Chief Executive Officer of the Wright Center for Graduate Medical Education and the Program Director for the Internal Medicine residency training program.

## Author Contributions

- Linda Thomas-Hemak conceived and designed the experiments, performed the experiments, analyzed the data, contributed reagents/materials/analysis tools, wrote the paper, prepared figures and/or tables, reviewed drafts of the paper, language editing.
- Ghanshyam Palamaner Subash Shantha conceived and designed the experiments, performed the experiments, analyzed the data, contributed reagents/materials/analysis tools, wrote the paper, prepared figures and/or tables, reviewed drafts of the paper, figures.
- Lakshmi Rani Gollamudi analyzed the data, contributed reagents/materials/analysis tools, wrote the paper, prepared figures and/or tables, reviewed drafts of the paper.
- Jignesh Sheth performed the experiments, analyzed the data, contributed reagents/materials/analysis tools, wrote the paper, prepared figures and/or tables, reviewed drafts of the paper.
- Brian Ebersole performed the experiments, contributed reagents/materials/analysis tools, wrote the paper, prepared figures and/or tables, reviewed drafts of the paper.
- Katlyn J. Gardner, Julie Nardella and Meaghan P. Ruddy contributed reagents/materials/analysis tools, wrote the paper, reviewed drafts of the paper.
- Lauren Meade conceived and designed the experiments, performed the experiments, analyzed the data, contributed reagents/materials/analysis tools, wrote the paper, reviewed drafts of the paper.

## Human Ethics

The following information was supplied relating to ethical approvals (i.e., approving body and any reference numbers):

Ethical approval was waived after review by the Institutional Review Board of the Wright Center for Graduate Medical Education, Scranton, PA on the 17th September 2013.

## Supplemental Information

Supplemental information for this article can be found online at http://dx.doi.org/10.7717/peerj.766#supplemental-information.

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
