# Peer review of "Nurturing 21st century physician knowledge, skills and attitudes with medical home innovations: the Wright Center for Graduate Medical Education teaching health center curriculum experience"

_PeerJ, doi:10.7717/peerj.766_

## Round 0.1 · original submission · Major Revisions

· Academic Editor

Major Revisions

This is an interesting study. Unfortunately I have to agree with the second reviewer that the use of resident self-assessments of their knowledge and skills is not in itself an adequate evaluation of the program. As noted in the manuscript at the bottom of page 14,

"Correlation of residents’ self-reported KSA of PCMH competencies with competency-based global 360 evaluations, direct observation tools, and ACGME reportable milestones outcomes, as well as patient care delivery process and health outcomes, are required before firm inferences can be made from our results."

I couldn't agree more. The study design is fairly robust and evaluating the effectiveness of these educational innovations would be of considerable interest. Just knowing residents feel they have gained knowledge and skills over the course of their educational program however is of limited value. Given PeerJ's review criteria I think the paper is publishable but it would be a lot more useful if it were possible to include measures of the residents' actual rather than self-reported knowledge and skills.

As noted by the reviewers, the PCMH Clinician Assessments are not described in adequate detail. It would be helpful to be able to view the scale items. Perhaps a copy could be placed on the Web with a link included in the Methods Section. Though the paper states the scale has been validated, no validity data or references to studies assessing the validity are provided other than scale reliability which in of itself is not a measure of validity. I think this issue needs to be addressed before the manuscript can be published.

It would also be helpful to have a better understanding of the PCMH curriculum. It's hard to make sense of the study results without knowing more about how the curriculum received by the THC residents differed from the TR residents. I expect the curriculum is described in detail somewhere on your program web site. A link to such a description would be useful and avoid having to lengthen the manuscript.

Please be sure to define each acronym used. A few were missed and may not be familiar to some readers, particularly those outside the USA. As noted by the second reviewer the writing could be be simplified. The use of so many acronyms even when defined makes it very hard to read if you are not familiar with the terminology. If there is a way to cut down on the use of acronyms, the paper would be easier to read. Also PeerJ does not provide copy editing so please check your revision very carefully for typos.

The two reviewers were kind enough to provide a considerable amount of additional feedback I hope you will find their feedback helpful in revising the manuscript.

·

Basic reporting

Very well-written article. Onequestion I had, which was not clearly stated, is:

It sounds as though the THC program was not developed until 2011? The article says " In 2011, THCGME pioneering grantees included 9 Family Medicine, one dental, and one Internal Medicine residencies." But it does not expressly state that their THC track started in 2011. If a THC program did in fact exist before implementation of the PCMH curriculum in 2011, it would have been helpful to have comparison surveys from 2009 and 2010 THC graduates (prior to implementation of the PCMH curriculum).

Figure 1 should have the top table labeled "May 2012" and the bottom table labeled "Nov 2012." Tables 1 - 4 are sufficiently labeled and easy to follow.

Experimental design

What is the PCMH Clinician Assessment tool? Is this a competency assessment created by the authors, or is it a validated / otherwise proven instrument used elsewhere and / or cited in the literature?

Validity of the findings

No comments

Additional comments

Overall, a very well-written paper, and exciting! I would definitely reference where the PCMH Clinicians Assessment tool can be found / used, as I would be interested in using it on residents at out institution!

Reviewer 2 ·

Basic reporting

The purpose of this prospective cohort study was to assess impact of patient centered medical home curriculum interventions on 24 Internal Medicine residents’ self-reported knowledge, skills and attitudes. Residents were divided into two groups; Traditional track and Teaching Health Center track, and follow during their three years of training. Curricular innovations were implemented July 2011 through May 2012 for the THC residents and then in 2012 were implemented across both programs. Semi-annual validated assessments of PCMH competencies were completed by all residents.

Introduction – The authors make a compelling case for the study since little literature is available on resident knowledge, attitude and skills related to PCMH competencies. Recent relevant literature is cited. The theoretical framework for curriculum design is not clearly presented.

Experimental design

Methods – The study was IRB approved (human subjects approval). The interventions were tested on one group of residents in the THC track before being applied to the other group in the TR track. Baseline data were captured on both groups of residents. Demographic details among the residents in both tracks were compared. Residents in early graduating classes also completed the assessments so comparisons could be made among residents receiving none, some and all of the PCMH interventions.

Although it is stated that “validated PCMH Clinician Assessments were completed by all Internal Medicine residents,” no evidence for validity is provided. Were they validated by the researchers or adopted from another source? Which PCMH competencies were knowledge, which were attitudes and which were skills? Explanation of the assessment tools is quite vague.

Validity of the findings

Results – Although the results are clearly presented and the tables and figures add value to data interpretation, it is difficult to assess the contribution of these data without knowing more about, or seeing, the self-assessment that residents completed. While an important contribution, self-assessment as a sole source of information is a low level program evaluation.

Discussion – On page 12 it is noted that “Our study provides evidence that intentional PCMH curricular innovations can improve residents’ self-reported KSA in areas reported as deficient in Kaiser Permanente’s survey.” What the Kaiser Permanente survey assessment applied in this study? Or, is this just a reference to PCMH deficiencies in general?

The limitations of this study are significant. Studying self-reported data as a sole source of assessment does not add value to the understanding of PCMH competency acquisition. As the authors note: “There remains a need to explore the relationship between enhanced residents’ self–reported KSA with milestones guided, competency based evaluations and direct observation tools as required by the ACGME’s Next Accreditation System, as well as with enhanced patient care delivery processes and health outcomes.” At least one other data source should have been included in the present study to lend credibility to the project. Including another source of data would have reduced some sources of bias. It is a good thing that the residents thought they gained in KAS but these data are not sufficient from a program evaluation perspective.

Additional comments

Grammatical revisions: In general, the wording of several sentences is overly complex, which makes for cumbrous reading. The manuscript needs editing for “ease of readability.” Examples:
1. This study provides evidence of changes in residents’ selfreported [self-reported should be hyphenated] KSA of PCMH competencies in support of the teaching health center based curriculum that aligns graduate medical education with the complexities of health care reform and patient care needs.
2. PCMH based didactics enhancements and resident engagement in reflective CQI within ambulatory venues with or pursuing NCQA PCMH certification were focally applied to 2011 THC residents from training initiation in July 2011 and were spread program wide in May 2012.
3. CQI projects were developed and shared as part of Pennsylvania Academy of Family Practice’s Improving Performance in Practice Residency Program/Community Health Center Collaborative with monthly team calls and quarterly learning sessions.

Sentence example #1 and #2 seem to be two sentences crammed together. Sentence #3 needs to be rearranged, e.g., CQI projects were developed and shared during monthly team calls and quarterly learning sessions as part of Pennsylvania Academy of Family Practice’s Improving Performance in Practice Residency Program/Community Health Center Collaborative. OR, “As part of Pennsylvania Academy of Family Practice’s Improving Performance in Practice Residency Program/Community Health Center Collaborative, CQI projects were developed and shared during monthly team calls and quarterly learning sessions.

Page 5; spell out FQHC the first time
Page 6; spell out CQI the first time

---

## Round 0.2 · accepted · Accept

· Academic Editor

Accept

Thank you for addressing the reviewers' and my concerns. I am pleased to accept your paper for publication.